# Infective prey leads to a partial role reversal in a predator-prey interaction

Veijo Kaitala[1]ᵒ*, Mikko Koivu-Jolma[2]ᵒ, Jouni Laakso[1]

1 Organismal and Evolutionary Biology Research Programme, Faculty of Biological and Environmental Sciences, Helsinki University, Helsinki, Finland, 2 Department of Physics, Faculty of Science, Helsinki University, Helsinki, Finland

ᵒ These authors contributed equally to this work.
* Veijo.Kaitala@Helsinki.Fi

**Data Availability Statement:** All relevant data are within the manuscript.

**Funding:** The authors received no specific funding for this work.

## Abstract

An infective prey has the potential to infect, kill and consume its predator. Such a prey-predator relationship fundamentally differs from the predator-prey interaction because the prey can directly profit from the predator as a growth resource. Here we present a population dynamics model of partial role reversal in the predator-prey interaction of two species, the bottom dwelling marine deposit feeder sea cucumber *Apostichopus japonicus* and an important food source for the sea cucumber but potentially infective bacterium *Vibrio splendidus*. We analyse the effects of different parameters, e.g. infectivity and grazing rate, on the population sizes. We show that relative population sizes of the sea cucumber and *V. Splendidus* may switch with increasing infectivity. We also show that in the partial role reversal interaction the infective prey may benefit from the presence of the predator such that the population size may exceed the value of the carrying capacity of the prey in the absence of the predator. We also analysed the conditions for species extinction. The extinction of the prey, *V. splendidus*, may occur when its growth rate is low, or in the absence of infectivity. The extinction of the predator, *A. japonicus*, may follow if either the infectivity of the prey is high or a moderately infective prey is abundant. We conclude that partial role reversal is an undervalued subject in predator-prey studies.

## Introduction

The ability of a prey to utilize the predator as a food source is referred to as a role reversal in predator-prey interaction [1–3]. The prey may become an enemy to the predator. Role reversal has been reported in cases, where the reversal coincides with age-depended size categories [1], age-depended susceptibility to predation [2], or changes in relative population abundances [3]. These role reversals are complete and include the idea that at some age or size a species switches being predated to being a predator itself. For example, the pike may be subject to predation by sticklebacks when it is small, but it develops into a real predator with increasing size.

The role reversal is incomplete or partial when the predator continues to hunt the prey while becoming or remaining vulnerable to the predation itself. In partial role reversal the

**Competing interests:** The authors have declared that no competing interests exist.

growth of the prey population relies on the prey's growth rate and on the additional resource acquiring by the infectivity, in particular, by its efficiency in killing and converting the predator into nutrition.

The sea cucumber (*Apostichopus japonicus*) is a bottom dwelling marine deposit feeder that uses its tentacled mouth to consume the topmost sediment layer [4, 5]. The sediment contains plant and animal debris, protozoa, diatoms and a diverse selection of bacteria [6–10]. Because the bacteria are abundant and have higher nutritional value than the surrounding sediment, they are considered a direct food source for detrivorous holothurians [11–13]. Specifically, bacteria are important dietary component of the sea cucumber *Apostichopus japonicus* [14]. Additionally, the bacteria that are not digested may have important function in providing the sea cucumber with essential nutrients that are not otherwise available [13, 15–18]. Furthermore, the bacteria in the diet of sea cucumbers is discussed in the research of Navarro et al. [19] and the references therein.

Bottom sediments are also inhabited by the opportunistic, potentially infective bacterium *Vibrio splendidus* [10, 20]. *V. splendidus* is an efficient decomposer, which allows it to thrive in many environments [21, 22]. To this end, *Vibrios* form a notable fraction of the bacterial flora in the sediment. Especially in nutrient rich areas near the range of *A. japonicus Vibrios* form one of the most abundant bacterial groups [22, 23]. Bacteria are most abundant in the detritus, where the bacterial density can be hundredfold compared to the sea water above the detritus [20, 23]. Because bacteria form an important food source for the sea cucumber, *A. japonicus* can be treated as a predator to *V. splendidus*. Consequently, *V. splendidus* is here considered as a prey for the generalist predator *A. japonicus* [5].

*V. splendidus* has been associated with seasonal epidemics of high mortality among the cultured sea cucumbers [24, 25]. On the other hand, *V. splendidus* can also survive in the gut of susceptible sea cucumbers [26, 27]. The interaction is not tight in the sense of traditional Lotka-Volterra predator-prey interaction since both species can also consume other resources.

We address the problem of partial role reversal in the predator-prey interaction by presenting a predator-infective prey model to analyse the dynamics and coexistence of the species. After presenting the basic framework of the model we parameterize the model for an opportunistic pathogenic bacteria and the commercially cultivated sea cucumber, an economically important species in aquaculture. The sea cucumber is appreciated as a delicacy and aphrodisiac widely in Asia. Even though the catches from the wild populations have drastically declined, the production of cultured sea cucumbers in year 2014 was over 200000 tonnes in China alone [28]. According to our results, the species most likely coexist at a stable equilibrium, but the infective prey can cause severe losses to the predator population. We also analysed the conditions for species extinctions. For the predator, the extinction depends on the infectivity of the prey, its population size as well as the grazing rate of the predator. The possibility of recognizing the presence and effects of an infective prey within a food web is of significant scientific and economic importance.

The variety of the antagonistic interactions between two species is substantial and it is worth of comparing our predator–infectious prey model with the more traditional models. Apart of the predator-prey interaction [29–31], the most important of the theories deal with disease [32, 33] and parasitoid models [34, 35]. The disease models often emphasize the way of transmission [36]. However, the number of species involved explicitly in the conceptual framework is also of central interest. The most common disease classes are direct transmission from host to host [37–39], parasitism [31, 34, 40] and vector mediated transmission [41, 42]. As any of these disease models may bear some resemblance with our predator–infectious prey model we address the similarities and differences in more detail in the Discussion.

## Modelling partial role reversal in predator-prey interaction

Let C, S and I denote the abundances of the prey, susceptible predator and infected predator populations, respectively. The generalist predator population (S) grows by grazing the sediment, which hosts the *Vibrio* prey (C) (Fig 1). Both species also use other resources for growth,

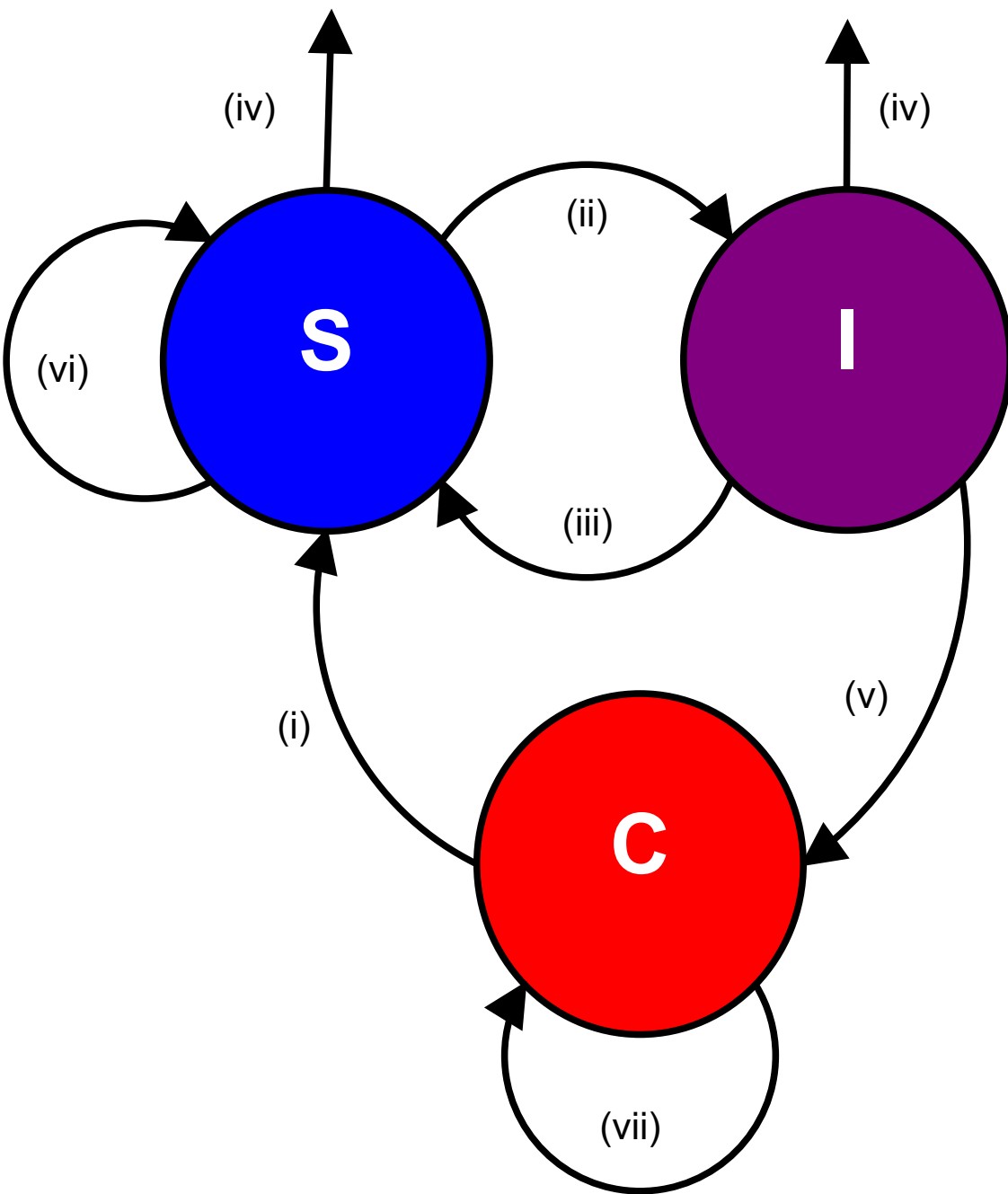

**Fig 1. A schematic presentation of the predator-infective prey model.** The predator (S), *A. japonicus*, is a bottom feeder (i). An essential part of the detritus hosts the parasite *V. splendidus* (C). Both *A. japonicus* and *V. splendidus* are generalists (vi, vii) such that they are able to survive in the absence of the other species. *V. splendidus* infects *A. japonicus* (I, ii). The disease mortality of *A. Japonicus* enhances the growth of the *V. splendidus* (v). A fraction of the infected *A. japonicus* recovers from the infection (iii). Both the healthy (S) and infected (I) hosts die naturally (iv).

meaning they are generalists or polyphagous and that each of them can survive as a single species population. As the prey is also pathogenic to the predator, a part of the predator population is infected, increasing the population size of infected predators (I). An infected predator can recuperate returning back to susceptible predators, or die naturally. It can also die of the infection by the *Vibrio* prey, thus, increasing the growth of the pathogenic prey. Unlike in SIR-type models, an infected predator does not infect susceptible predators. Infection occurs only through the harvesting infective prey. A significant aspect in our predator-prey interaction is that both the prey and the predator are only a part of a food web. Both species have a base growth rate that is independent of their mutual interaction, and they both are able to grow independently according to the respective carrying capacity of the environment.

The differential equation model for the prey-predator dynamics is given as

$$\frac{dC}{dt} = r_C C\left(1 - \frac{C}{K_C}\right) - CgS + e_{IC}\mu_{inf}I \tag{1}$$

$$\frac{dS}{dt} = r_S S\left(1 - \frac{S}{K_S}\right) - e_{SI}\alpha CgS + (1-\alpha)e_{CS}CgS - \mu_S S + \beta I \tag{2}$$

$$\frac{dI}{dt} = e_{SI}\alpha CgS - I\left(\mu_S + \mu_{inf} + \beta\right), \tag{3}$$

where the increases of the prey and predator abundances are both defined by logistic growth terms. Parameters $r_i$ and $K_i$ are the growth rates and carrying capacities of the prey and predator, respectively. Parameter $g$ ($0 \leq g \leq 1$) denotes the grazing rate of the predator $S$ for prey $C$. This can be interpreted either as a fraction of the feeding area grazed during a time step or it can equally be interpreted as the prey selectivity coefficient of the predator. Thus, the total number of the prey harvested by the predator is $gCS$. Parameter $\alpha$ denotes the fraction of the infective prey from the total prey population. Thus, from the predation rate $CgS$ fraction $(1-\alpha)$ increases the growth rate of the susceptible predator population with a prey to predator conversion efficiency $e_{CS}$. The rest of the harvested prey, $\alpha CgS$, infects the predator population at infection rate $e_{SI} > 0$, converting a corresponding fraction of the susceptible predator population to infected. Parameter $\beta$ denotes the recovery rate of the infected predators. Parameter $\mu_{inf}$ denotes the predator infection mortality and parameter $e_{IC}$ predator to prey conversion efficiency. Finally, $\mu_S$ denotes the natural predator mortality.

## Parametrization of the model

Most of the parameters were chosen according to the corresponding values available in the literature. Conversion efficiencies were calculated using the assimilation efficiency and the dry weights of the predator and the prey. The efficiency of digestion of bacteria in the sea cucumber gut has been estimated to range between 40% [12, 15] and 80% [13]. However, Plotieau et al. [43] showed that γ-Proteobacterial genus, of which *Vibrio spp.* are part of, are digested with lower efficiency. Therefore, assimilation efficiency was set to conservative value of 0.25 for the sea cucumber. For the bacteria, assimilation efficiency was set to 0.5 [44]. The dry weight of the bacterium *V. splendidus* varies from 145fg in stationary phase cells to 850fg in exponential phase cells [45]. In the simulations we used 280fg as the dry weight, which coincides with the dry weight of another common bacterium of similar size E. coli [46]. The dry weight of *A. japonicus* is calculated according to Sun et al. [4] who reported that the dry weight of *A. japonicus* equals 0.075×wet weight. The mean wet weight $m_{Aj}$ was set at 150g [47].

Prey to predator conversion efficiency is calculated as $e_{CS} = \frac{m_{bact}}{m_{Aj}}0.25 = \frac{0.28 \cdot 10^{-12}g}{150g \cdot 0.075}0.25 = 6.22 \cdot 10^{-15}$.

Predator to prey conversion efficiency is $e_{IC} = \frac{m_{A.j}}{m_{bact}} 0.5 = \frac{150g \cdot 0.075}{0.28 \cdot 10^{-12}g} 0.5 = 1.00 \cdot 10^{13}$.

Because the prey is an opportunistic bacteria, it is distributed within the detritus. The slow moving predator attacks the prey by grazing. The prey population available to the predator depends on the population density of the prey in the detritus and the volume of detritus available to the predator. The grazing area is limited by the population density of the predator.

According to the empirical study by Lysenko et al. [47] the natural population density of *A. japonicus* is 0.14 individuals per square meter. The population density at the carrying capacity of the bacteria $K_C$ was taken from the literature [7, 20]. Therefore, the area of the feeding unit is set to $A_{KS} = \frac{1m^2}{0.14} = 7m^2$ and depth of foraging to $1cm$. We calculated the predator grazing rate using the formula

$$g = \frac{m_{Aj}f}{\rho V} = \frac{150g \cdot 5.3 \cdot 10^{-3}g^{-1}h^{-1}mg}{1gcm^{-3} \cdot 70000cm^2 \cdot 1cm} = 1.14 \cdot 10^{-5}h^{-1} = 2.73 \cdot 10^{-4}d^{-1}.$$

where $m_{Aj}$ is the wet weight of the sea cucumber and $f$ is the amount of sediment eaten by the sea cucumber per hour per gram of sea cucumber [4], $\rho$ is the density of the sediment as given by Kennish [9], and $V$ is the volume of the feeding unit. The resulting grazing rate is the portion of the available prey eaten within a time step. Because the actual grazing rate also depends on the selectivity of the predator [43, 48], a range of grazing rate values around the nominal value was used in model analysis and numerical simulations. For the carrying capacity of the sea cucumber we used $K_S$ = s 10000, which corresponds to a grazing area of $70000m^2$ and the carrying capacity of *V. spendidus* $K_C = 1 \cdot 10^{13}$.

The symbols and parameters used in the model and are shown in Table 1.

## Model analyses

### Population equilibria

The equilibrium of the community is the starting point of the analysis of community behaviour. The equilibrium is defined by assuming the time derivatives in the population Eqs (1)–(3) equal to zero.

**Table 1. Symbols and parameter values.**

| Parameter | | Value | Unit |
|---|---|---|---|
| Susceptible predator | S | *A. japonicus*, | |
| Infected predator | I | *A. japonicus*, | |
| Infective prey | C | *V. splendidus*, | |
| Infective prey growth rate | $r_c$ | 0.5, 5.0, 50 | $d^{-1}$ |
| Susceptible predator growth rate | $r_s$ | 0.02 | $d^{-1}$ |
| Prey K | $K_c$ | $1 \cdot 10^{13}$ | |
| Predator K | $K_s$ | 10000 | |
| Predator infection mortality due to the prey | $\mu_{inf}$ | 0.8 | $d^{-1}$ |
| Predator mortality | $\mu_S$ | 0.01 | $d^{-1}$ |
| Predator grazing rate | g | $1.0 \cdot 10^{-11} - 10.0 \cdot 10^{-4}$ | $d^{-1}$ |
| Prey to predator conversion efficiency | $e_{CS}$ | $6.22 \cdot 10^{-15}$ | |
| Predator to prey conversion efficiency | $e_{IC}$ | $1.0 \cdot 10^{13}$ | |
| Infectivity of the prey | $e_{SI}$ | $10^{-13} - 10^{-9}$ | |
| Proportion of infective prey | $\alpha$ | 0.001–1.0 | |
| Infected predator recovery | $\beta$ | 0.2 | $d^{-1}$ |

Let us denote

$$z = e_{SI}\alpha g/(\mu_S + \mu_{inf} + \beta).$$ (4)

From Eq (3) we get $I = zCS$. Inserting $I$ into Eqs (1) and (2) and dividing the resulting equations by C and S, respectively, we get

$$0 = r_C\left(1 - \frac{C}{K_C}\right) - gS + e_{IC}\mu_{inf}zS$$ (5)

$$0 = r_S\left(1 - \frac{S}{K_S}\right) - e_{SI}\alpha Cg + e_{CS}(1 - \alpha)Cg - \mu_S + \beta zC$$ (6)

Linear Eqs (5) and (6) can be presented in a matrix form

$$A\begin{bmatrix} C \\ S \end{bmatrix} = \begin{bmatrix} r_C \\ r_S - \mu_S \end{bmatrix},$$ (7)

where

$$A = \begin{bmatrix} a_{11} & a_{12} \\ a_{21} & a_{22} \end{bmatrix} = \begin{bmatrix} \dfrac{r_C}{K_C} & g - e_{IC}\mu_{inf}z \\ e_{SI}\alpha g - e_{CS}(1 - \alpha)g - \beta z & \dfrac{r_S}{K_S} \end{bmatrix}$$ (8)

The solution of Eqs (5) and (6) is given as

$$\acute{C} = \frac{1}{a_{11}a_{22} - a_{12}a_{21}}(a_{22}r_C - a_{12}(r_S - \mu_S))$$ (9)

$$\acute{S} = \frac{1}{a_{11}a_{22} - a_{12}a_{21}}(-a_{21}r_C + a_{11}(r_S - \mu_S))$$ (10)

Recall that the population size of infected predators are then calculated as $\acute{I} = z\acute{C}\acute{S}$ where z is defined by Eq (4).

The equilibrium states of interest are the following:

a. Both species coexists at a general equilibrium: $\acute{C}, \acute{S}, \acute{I} > 0$. Stability of this equilibrium represent continuing coexistence of the species.
   In the absence of species interaction the carrying capacity of the prey is equal to $K_C$ and that of the predator is equal to $\frac{(r_S - \mu_S)K_S}{r_S}$.

b. Infective prey exists but is zero: $\acute{C} = 0, \acute{S} > 0, \acute{I} = 0$. This represent an extinction of the prey.

c. Predator is absent: $\acute{C} > 0, \acute{S} = \acute{I} = 0$. This represents an extinction of the predator.

Two coexistence equilibria with interacting species are of special interest. First, if we have in Eq (8)

$$a_{12} = g - e_{IC}\mu_{inf}e_{SI}\alpha g/(\mu_S + \mu_{inf} + \beta) = 0$$ (11)

then the equilibrium population size of the prey is $\acute{C} = K_C$. Condition (11) also yields an

equilibrium population size of the predator equal to

$$\acute{S} = -K_C(e_{SI}\alpha g - e_{CS}(1-\alpha)g - \beta z) + \frac{K_S(r_S - \mu_S)}{r_S}.$$

This outcome is referred to as the *switch point of the prey growth rate* because condition (11) neutralizes the effect of the prey growth rate on the equilibrium population sizes of both species. Condition (11) is independent of the grazing rate of the predator, but any increase (decrease) in the infection-related parameters $e_{IC}$, $e_{SI}$ and $\alpha$ will make $a_{12} < (>)$ 0. However, the ultimate effect of the parameter change on the population size of the prey also depends on the effect of these parameters in $a_{21}$.

A similar analysis with respect to the equilibrium population size of the predator is valid. If

$$a_{21} = e_{SI}\alpha g - e_{CS}(1-\alpha)g - \beta e_{SI}\alpha g/(\mu_S + \mu_{inf} + \beta) = 0 \tag{12}$$

In this case, $\acute{S} = \frac{(r_S - \mu_S)K_S}{r_S}$. Again, the effect of the prey growth rate is neutralized in the equilibrium solution of the predator. However, condition (12) does not have a comparable effect on the equilibrium population size of the prey. This outcome is referred to as the secondary *switch point*. Again, condition (12) is independent of the grazing rate of the predator. However, it is dependent of the same parameters as condition (11). When conditions (11) and (12) do not hold then the prey growth rate is not neutralized, in which case it affects the equilibrium population sizes. These switch points are illustrated in the Results.

## Numerical simulations

The numerical simulations of the model (1)-(3) were performed using Matlab R2020b. Numerical simulations were in accordance with the analytical results (Section "Population equilibria"). The effects of infectivity of the prey $e_{SI}$, proportion of the infective prey $\alpha$, and the grazing rate $g$ were tested using wide parameter ranges. Though the outbreaks caused by *V. splendidus* have been associated with high mortality rates, we tested the model also with low infection mortality rates and high recovery rates.

## Results

For a prey with a high growth rate $r_C$ the level of infectivity, $e_{SI}$, does not crucially affect the prey population size (Fig 2). The prey population size will settle around the level of carrying capacity $K_C$. For low infectivity $e_{SI}$ a higher prey growth rate $r_C$ supports larger prey population than a lower growth rate, but this is reversed when $e_{SI}$ increases beyond the switch point of the prey growth rate (o, Eq (11)), where high and low growth rates of the prey provide equal population sizes. An increase in infectivity $e_{SI}$ tend to result in greater prey and lesser predator population sizes. However, if the infectivity is high enough and the growth rate is low the trend of the prey population size turns into decreasing. Low infectivity leads to the extinction of the slowly growing prey.

A low infectivity $e_{SI}$ combined with a high growth rate $r_C$ of the prey can be beneficial also for the predator because the predator is able to sustain population levels above the carrying capacity $\frac{(r_S - \mu_S)K_S}{r_S}$. Rising the level of infectivity, however, decreases the predator population. At the extinction of the prey (at low infectivity values and low prey growth rate) the predator population size settles down at its carrying capacity. However, when the prey growth rate is higher allowing the prey to survive at low infectivity rates then the population sizes of the predator may surpass the level of the carrying capacity. The secondary switch point (Δ, Eq (12)) provides a threshold for the infectivity below which the population size of the predator exceeds the carrying capacity.

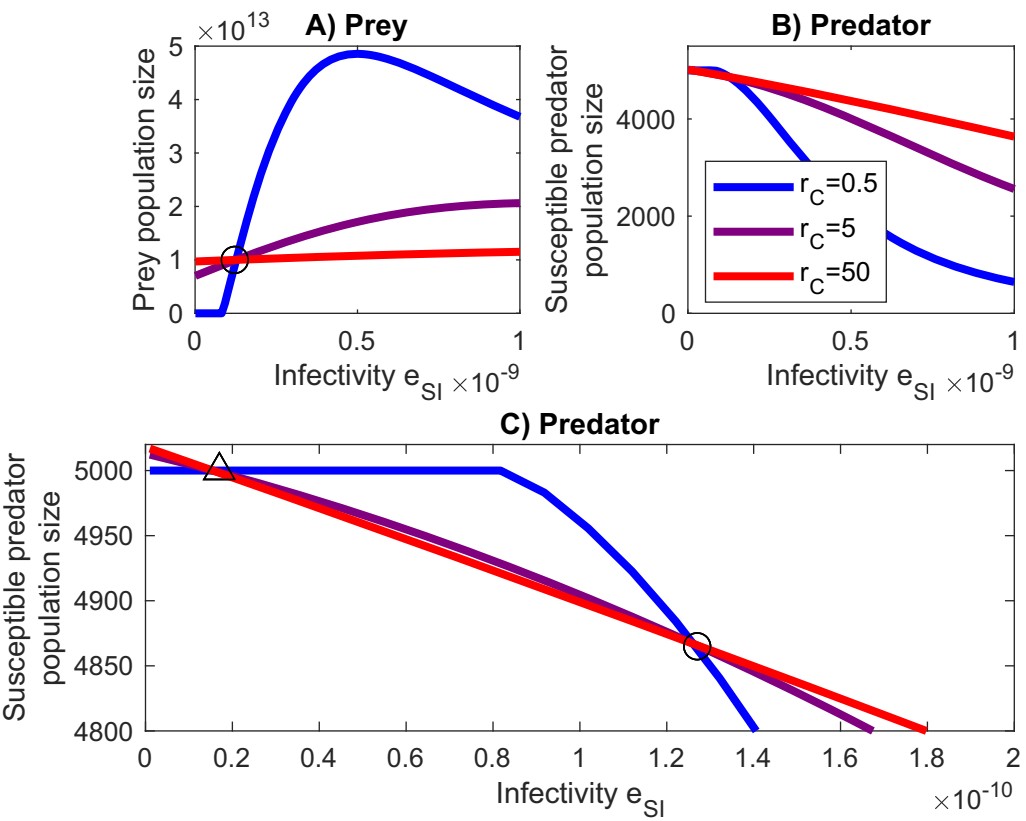

**Fig 2. Infectivity affects both prey and predator population sizes.** (A) and (B) The population size of the prey and predators, respectively. (C) A closer view to the switch points (o, △) in the predator population. Red, purple and blue lines represent fast ($r_C = 50$), medium ($r_C = 5$) and slow ($r_C = 0.5$) growth rates. The predator's grazing rate $g = 3.0 \cdot 10^{-4}$ and the infective proportion of the prey $\alpha = 0.001$.

High infectivity $e_{SI}$ increases the population size of a slowly growing prey because increasing infectivity allows the prey to reach a higher prey population size as compared to a prey with a higher growth rate $r_C$ (Fig 2). However, due to the high mortality $\mu_{inf}$ associated with the infection, a too high infectivity causes the extinction of the predator and a decline in the prey population size Fig 2 also illustrates the presence of a switch point such that the relative population sizes will change with the change of a parameter. When $e_{SI}$ = 1.22e-10, making $a_{12}$ = 0 (Eq 11), the equilibrium population size of the prey equals its carrying capacity $\acute{C} = K_C$. At the same value of infectivity the equilibrium population size of the predator will be $\acute{S} = 4865$. Below the switch point slow prey growth rates supports lower prey population sizes than higher growth rates. When the parameter $e_{SI}$ passes the switch point then the relative population sizes are reversed. The effect of the switch point to the population sizes of the predator is opposite. Note that the switch point is always related to a chosen parameter. Condition $a_{12} = 0$ can become true by choosing appropriate combinations of parameter values for $e_{SI}$, $\alpha$, $e_{IC}$ and $\mu_{inf}$. A comparable analysis can be carried out for the solutions for the condition $a_{21} = 0$ (Eq 7). For a set of parameter values making $a_{21} = 0$ the equilibrium population size of the predator would equal to its carrying capacity $\acute{S} = \frac{r_S - \mu_S}{r_S} K_S$. For parameter values with $a_{21} \neq 0$ the equilibrium values of the predator would depend on the specific parameter values.

Infectivity $e_{SI}$ and the proportion of infective prey in the total prey population $\alpha$ have parallel but not completely interchangeable effects on the population sizes of the prey and predator (Fig 3).

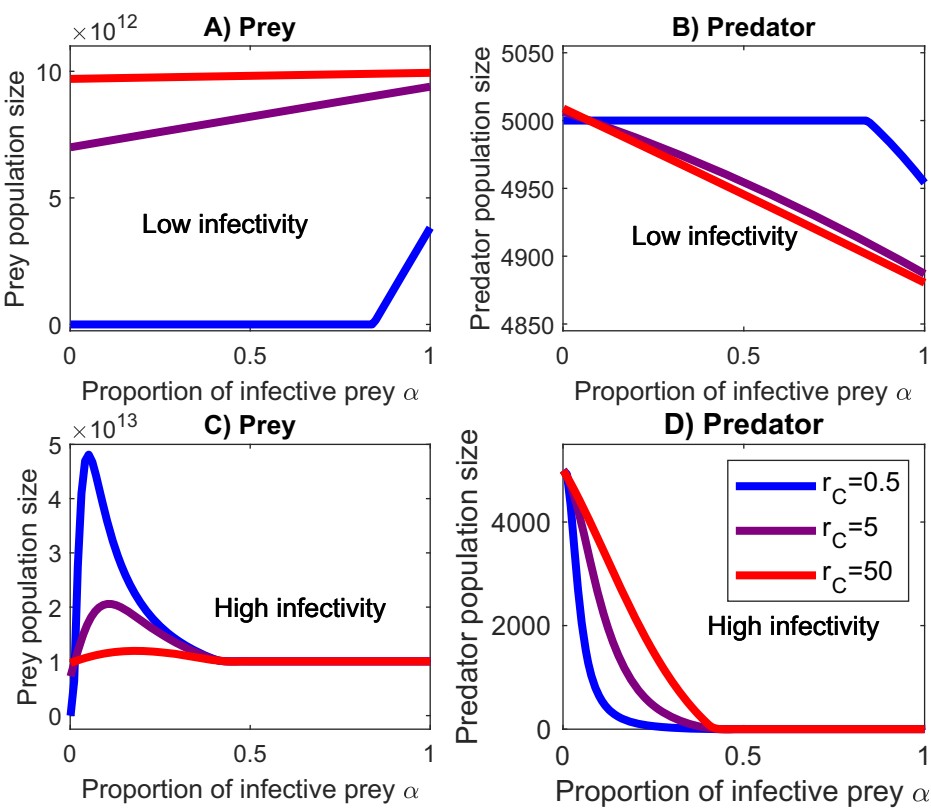

**Fig 3. Proportion of infective prey $\alpha$ affects the population in the same way as infectivity $e_{SI}$.** (A) Slowly growing prey with low infectivity can proliferate only if the majority of the prey are infective. (B) Even high proportions of infective prey cause only a slight decrease in predator population. High infectivity $e_{SI}$ prevents the extinction of the prey. (C) Highly infective prey thrives best when it forms relatively small part of the prey population because (D) the predator becomes extinct if the majority of the prey are infective ($\alpha \gtrsim 0.4$). Red, purple and blue lines are fast ($r_C = 50$), medium ($r_C = 5$) and slow ($r_C = 0.5$) growth rates. Infectivity values are $e_{SI} = 10^{-13}$ in subfigures A and B, and $e_{SI} = 10^{-11}$ in subfigures C and D.

If the infectivity of the prey is very low ($e_{SI} = 10^{-13}$, Fig 3A and 3B), the predator will survive any proportion of the infective prey, and can even completely eradicate a slow growing prey. In contrast, if the infectivity is high ($e_{SI} = 10^{-11}$, Fig 3C and 3D), then the predator will become extinct even at relatively low infective prey densities. This happens regardless of the prey growth rate.

If grazing rate approaches zero both the prey and the predator population sizes tend towards the carrying capacity regardless of the infectivity (Fig 4). Increasing grazing rate may have different effects on the prey and predator sizes. When the value of the infectivity remains low, an increase in the grazing rate benefits the predator (Fig 4B). High growth rate of the prey results in larger predator population than low growth rate. If the value of the infectivity is increased slightly (moderate infectivity) an increment in growth rate decreases predator population sizes (Fig 4D). In both cases increasing grazing rate decreases the prey population size (Fig 4A and 4C).

## Extinction of the species

We consider here the possibility of extinction of the predator or the prey. The questions of interest are: 1) Under which conditions the predator can drive the prey to extinction such that

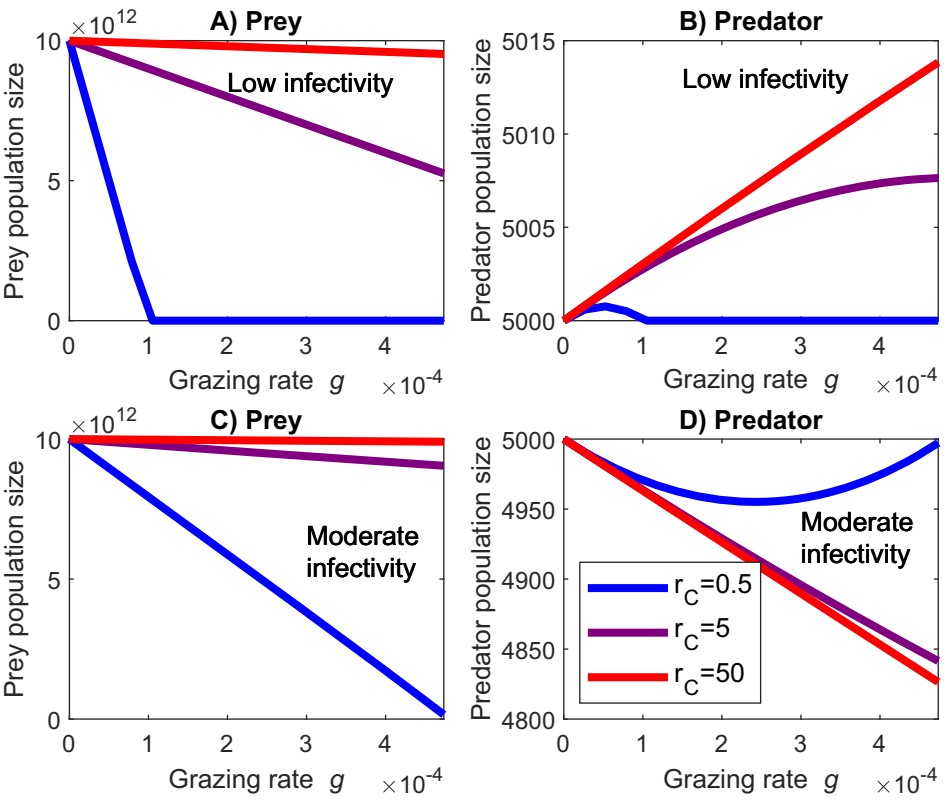

**Fig 4. An increase in the infectivity may reverse the effect of predator grazing rate on the predator population size.** (A) and (B) The prey's infectivity $e_{SI}$ is low. Increasing the predator grazing rate $g$ decreases prey and increases predator population sizes. (C) and (D) In contrast, if $e_{SI}$ is slightly greater, then increasing grazing rate decreases the population levels of the prey as well as of the predator. Red, purple and blue lines are fast ($r_C = 50$), medium ($r_C = 5$) and slow ($r_C = 0.5$) growth rates. The infectivity values are $e_{SI} = 10^{-13}$ in subfigures A and B, and $e_{SI} = 10^{-10}$ in subfigures C and D.

the species community would settle on the "predator only" equilibrium $\acute{C} = 0, \acute{S} > 0, \acute{I} = 0$. 2) Alternatively, we ask under what conditions the prey can eradicate the predator such that the species community would ultimately settle on the "prey only" equilibrium $\acute{C} > 0, \acute{S} = \acute{I} = 0$.

Consider first the "predator only" equilibrium $\acute{C} = 0, \acute{S} = \frac{r_S - \mu_S}{r_S}K_S, \acute{I} = 0$, that is, the sea cucumber lies at its carrying capacity and *V. splendidus* has been driven to extinction. If the equilibrium is locally stable then the extinction of *V. splendidus* is expected to occur. The local stability of the linearized dynamics at the equilibrium can be analysed studying the properties of the following Jacobian matrix

$$
J = \begin{bmatrix} J_{11} & J_{12} & J_{13} \\ J_{21} & J_{22} & J_{23} \\ J_{31} & J_{32} & J_{33} \end{bmatrix} = \begin{bmatrix} r_C - \dfrac{g(r_S - \mu_S)}{r_S}K_S & 0 & e_{IC}\mu_{inf} \\ g\dfrac{r_S - \mu_S}{r_S}K_S[e_{CS}(1-\alpha) - e_{SI}\alpha] & -r_S + \mu_S & \beta \\ ge_{SI}\alpha\dfrac{r_S - \mu_S}{r_S}K_S & 0 & -(\mu_S + \mu_{inf} + \beta) \end{bmatrix}
$$

Recall that if the real parts of the eigenvalues of J are all negative the system is locally stable. It can be shown that the first eigenvalue $\lambda_1 = -r_S + \mu_S$ is negative. The remaining two eigenvalues depend on the submatrix where line 2 and column 2 are deleted in matrix J. The eigenvalues $\lambda_2, \lambda_3$ both have negative real parts if and only if [49].

$$J_{11} + J_{33} = \left( r_C - g\frac{r_S - \mu_S}{r_S} K_S \right) - \left( \mu_S + \mu_{inf} + \beta \right) < 0$$

and

$$J_{11}J_{33} - J_{13}J_{31} = -\left( r_C - g\frac{r_S - \mu_S}{r_S} K_S \right)\left( \mu_S + \mu_{inf} + \beta \right) - e_{IC}\mu_{inf}ge_{SI}\alpha\frac{r_S - \mu_S}{r_S} K_S > 0$$

In this case extinction occurs. For example, if the proportion of infective prey is low ($\alpha \approx 0$) and the growth rate is low ($r_C < g\frac{r_S - \mu_S}{r_S} K_S$) then both conditions become true and the predator will eradicate the prey. High proportion of infective prey, high energetic efficiency and high carrying capacity may protect the prey from extinction.

The extinction of the prey depends crucially also on the grazing rate g of the predator (Fig 5). There is a threshold value or a minimum grazing rate g at which the predator can cause the extinction of the prey.

If the prey is a specialist such that it consumes only the predator ($r_C = 0$), then the extinction can be due to low infectivity or insufficient infective population. Yet, even a low prey growth

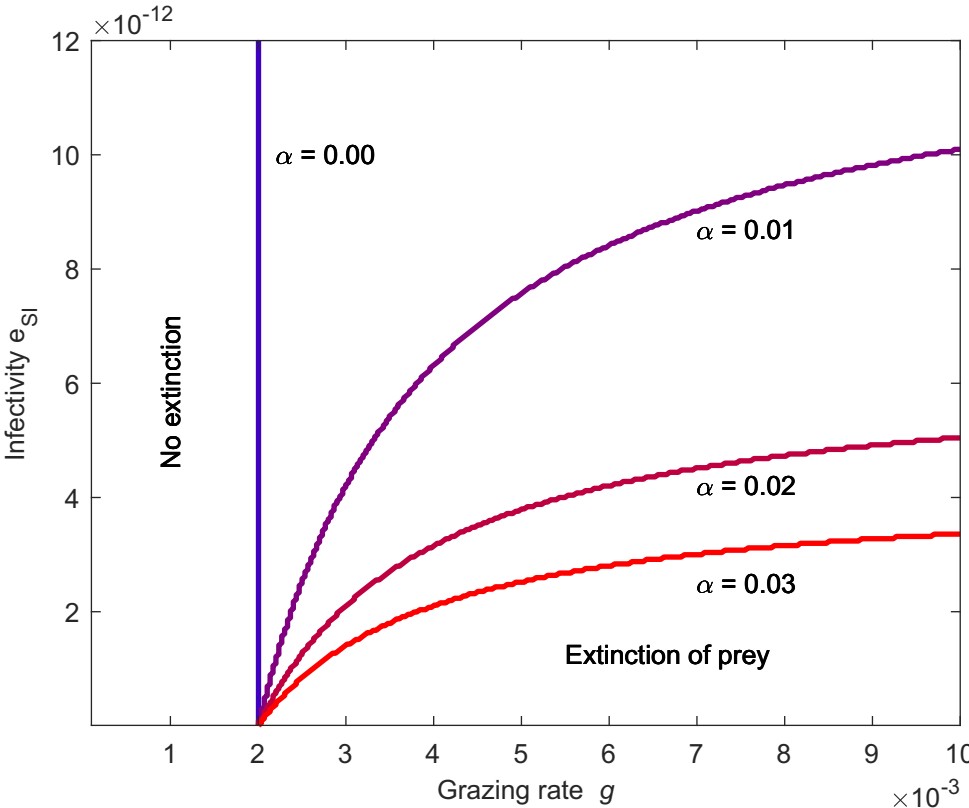

**Fig 5. Predator grazing rate g affects the extinction of prey.** Extinction of prey is possible if the predator's grazing rate g is higher than the threshold defined by the prey's growth rate. If the prey's infectivity $e_{SI}$ is high, only a small fraction $\alpha$ of the prey population needs to be infective to escape the extinction. Prey growth rate is $r_C = 10$.

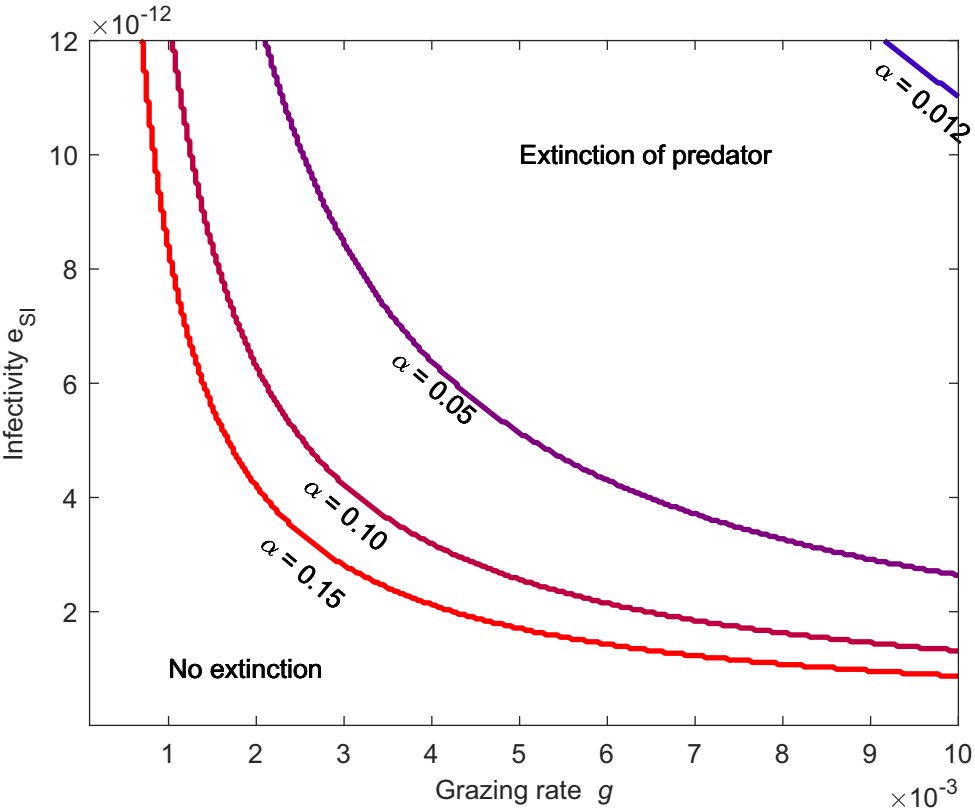

**Fig 6. The eradication of the predator by the prey is possible above the lines defining the fraction α of the infective prey.** If the infective prey forms a small fraction α of the available prey population, the high prey infectivity $e_{SI}$ and predator grazing rate g are needed to eradicate the predator. To the contrary, if the infective fraction α is large a low infectivity $e_{SI}$ or a low grazing rate g is needed for the predator to prevail. Using the parameters for *A. japonicus* and *V. splendidus* the predator survives always if α<0.011. At very low grazing rate or infectivity the predator does not consume enough infective prey to suffer extinction. Prey growth rate is $r_C = 10$.

rate may keep the prey population alive, if the prey is infective enough. Because infective prey can survive when its outside growth rate $r_C = 0$, a positive growth rate guarantees the survival also in the case of relatively low infectivity, if the infective prey forms large part of the predator's diet.

Extinction of the predator leads to the "prey only" equilibrium $\acute{C} = K_C, \acute{S} = I = 0$. The linearized dynamics of the predator-prey interaction at the equilibrium can be presented as

$$
J = \begin{bmatrix} J_{11} & J_{12} & J_{13} \\ J_{21} & J_{22} & J_{23} \\ J_{31} & J_{32} & J_{33} \end{bmatrix} = \begin{bmatrix} -r_C & -gK_C & e_{IC}\mu_{inf} \\ 0 & r_S - K_Cg(e_{SI}\alpha - e_{CS}(1-\alpha)) - \mu_S & \beta \\ 0 & ge_{SI}\alpha K_C & -(\mu_S + \mu_{inf} + \beta) \end{bmatrix}
$$

The first eigenvalue of the Jacobian matrix J is $\lambda_1 = -r_C$. The remaining two eigenvalues depend on the submatrix with line 1 and column 1 deleted in the Jacobian matrix J. The real parts of the eigenvalues $\lambda_2, \lambda_3$ are both negative if and only if [49]

$$
J_{22} + J_{33} = r_S - K_Cg(e_{SI}\alpha - e_{CS}(1-\alpha)) - \mu_S - (\mu_S + \mu_{inf} + \beta) < 0
$$

and

$$J_{22}J_{33} - J_{23}J_{32} = -[r_S - K_C g(e_{SI}\alpha - e_{CS}(1 - \alpha)) - \mu_S](\mu_S + \mu_{inf} + \beta) - \beta g e_{SI} \alpha K_C > 0$$

The proportion of the infective prey $\alpha$ is crucial. Assume that the prey is not infective, $\alpha = 0$. Then $J_{22}J_{33} - J_{23}J_{32} = -[r_S + K_C a e_{CS} - \mu_S](\mu_S + \mu_{inf} + \beta) < 0$ indicating that the predator does not become extinct. Assume next that $\alpha = 1$. Then $J_{22} + J_{33} < 0$ if

$$r_S - \mu_S - (\mu_S + \mu_{inf} + \beta) < K_C g e_{SI}$$

and $J_{22}J_{33} - J_{23}J_{32} > 0$ if $-K_C g e_{SI}(\mu_S + \mu_{inf}) - (r_S - \mu_S)(\mu_S + \mu_{inf} + \beta) > 0$, indicating that the predator will become extinct. Thus, in the case that the infectivity of the prey is high the extinction of the predator becomes true when the predator growth rate is low as compared to the background mortality $\mu_S$. When the infectivity of the prey is moderate the conditions for the extinction are affected in a nonlinear way by the growth rate of the predator, $r_S$, the carrying capacity of the prey, $K_C$, the grazing rate of the predator, $g$, and the infectivity of the prey, $e_{SI}$. Fig 6 presents how the change in infectivity $e_{SI}$ together with the different infective prey proportions $\alpha$ affect the extinction of predator. If the prey's growth rate $r_C$ is at all positive, the prey can drive the predator to extinction. Interestingly, if $r_C > 0$ the level of growth rate does not affect the results.

## Discussion and conclusions

We have presented a new predator-prey model with partial role reversal where the predator can become a target of infections by the prey such that the prey can use the predator as a resource for growth. In our model the prey and the predator both can utilize alternative resources and can grow in the absence of their respective prey or host.

We parametrized the model using sea cucumber *A. japonicus* and a bacterium species *V. splendidus* as a model system. Numerical results are also applicable to other opportunistic bacteria. For example, the parameters used for *V. splendidus* coincide with the parameters for E. coli [46]. Decomposer bacteria are abundant in the sediment [20, 22, 23], especially at the nutrient rich sea grass beds [23]. Because sea grass beds and the adjacent areas host also large numbers of detrivores and bacterivores that include sea cucumbers [23], bacteria play important part in the food web [11, 18], although the details of the interactions are still poorly understood.

According to the model, it can be beneficial for a slow growing opportunistic prey to have a vulnerable predator. In essence, if the prey can grow in the absence of the predator and it has the ability to infect, kill, and consume the predator, it benefits always from an increase in infectivity. Increasing infectivity increases also the prey population size at equilibrium, even above the carrying capacity. However, for an opportunist that has slow growth rate, the benefit from increasing infectivity is greater than for fast growing opportunist.

The switch point, where slow and fast growth rate result in equal population sizes, depends on the pathogenic qualities of the prey. If infectivity is lower than the threshold value at switch point, fast growing opportunist reaches larger population sizes. To the contrary, if infectivity is higher than the switch point value, slow growth rate results in larger prey population size. At switch point a small change in infectivity or mortality of the prey can select fast or slow growing prey as the most proliferous variant. In the case of single predator, this sets limits to the combination of infectivity and growth rate for an opportunistic prey.

We also analysed the conditions for species extinction. A generalist predator becomes eradicated if the prey population is infective enough, the infection can cause mortality, and the abundance of the infective prey is high. For a specialist predator the extinction depends on the infectivity of the prey, and its population size as well as the grazing rate of the predator. The

extinction of the prey is possible if its infectivity is low and either it the infective prey forms only a small fraction of total prey population or the growth rate is low.

The infective prey-predator model departs from an ordinary predator- prey model by quantifying the infectivity of the prey and the mortality of the infection. In principle, this resembles a fatal infectious disease. However, the predator is also able to consume the prey regardless of its pathogenicity, and can therefore benefit from the growing pathogenic prey population. It is noteworthy that infection does not need to be a bacterial infection. Any similar situation can play the part of a disease in the model framework. In these examples, the infectivity is taken as a number that describes the ability of the prey to find potential victims among the predators and to attract the rest of the prey population to the site. The infection mortality describes the probability of death of an infected or grazed predator.

It is enlightening to assess our model in the context of the enormous body of epidemiological and host-pathogen interactions [50]. One of the most famous disease categories is based on direct transmission of the parasites among the host individuals [36]. The traditional SIR models are compartment models that divide host individuals into the categories of susceptible, infectious, and recovered [32, 33]. They describe the reproduction and mortality of the host where the individuals become infected through contact infection, and die or recover from the disease. The variants of the host dynamics can vary depending, for example, on whether all the S-, I-, and R-individuals reproduce, or whether all or only a fraction of infected individuals subsequently either die, recover or become sterilized, and whether or not infected hosts have density-dependent interactions with healthy hosts. Nevertheless, the SIR-models do not describe any species interaction with the agent causing the disease.

Another equally significant model class is the host-parasitoid models, the theory of which was first presented by AJ Nicholson and VA Bailey [34]. The Nicholson–Bailey model is a discrete-time model that describes insect host–parasitoid interactions [51]. Both host and parasitoid have a single, nonoverlapping generation per year. Parasites are often host specific. Although most of the host parasitoid theory has been developed using discrete time models, continuous time models have been reported [52, 53].

A third class of disease models include disease transmitting vectors [42, 54]. More complicated life-cycles of the parasite may include an additional, intermediate host. If the intermediate host is a more or less passive transport vechicle to transmit a parasite from one host to another it is called a vector [36]. Intermediate hosts can also have other functions, e.g., overcoming adverse environmental conditions. Vector based transmission introduces at least one more organism, the vector, into the host-parasite interaction.

We consider yet another class of diseases, environmentally growing opportunist pathogens [55–62], which differ from obligatory pathogens in their ability to survive and replicate e.g. as saprotrophs in the outside-host environment. They are able to infect multicellular hosts and is which way they use within-host replication more of an alternative reproduction strategy [63–65]. Well-known examples of pathogens of this class are *Vibrio cholerae*, *Flavobacterium columnare*, and *Bacillus anthracis*, all of which cause sporadic outbreaks (for a review, see [66]). Environmental opportunist pathogens are not dependent on live hosts for transmission and the hosts do not consume the pathogens. By definition, they are polyphagous (generalists) and not as host specific as obligatory pathogens.

As a whole, the SIR models consider the disease dynamics only on a single trophic level with no reference to predation. Moreover, the transmissions occur through contacts. The host-parasitoid models resemble the predator-prey models. However, the parasitism is tightly considered to be unidirectional. Vector based spreading of a disease requires at least one additional intermediate host. Finally, the environmentally growing opportunist pathogens are not predated by the host such that the parasite or predatory relationship is unidirectional. We

emphasize this difference since they seem to have different dynamic properties. The environmentally growing opportunist pathogens produce easily unstable oscillating dynamics whereas in the presence of the predatory host no oscillations are observed. Overall, we conclude that predator-infective prey model differs from the other disease classes in some crucial points. We consider our study be an example case of role reversal interaction in a predator-prey interaction, even if the prey happens to be a parasite.

The system consisting of a predator and an infective prey remains mostly an unresearched subject. The population model presented here describes the process of role reversal using several parameters. However, of these parameters only a few were in the model markedly involved in the role reversal process. These include the prey to predator and predator to prey conversion efficiencies, and the infectivity of the prey. This implies that it would be possible to address the subject empirically by studying a suitable pair of model organisms.

## Acknowledgments

We thank Raine Kortet for useful discussions.

## Author Contributions

**Conceptualization:** Veijo Kaitala, Mikko Koivu-Jolma, Jouni Laakso.

**Data curation:** Mikko Koivu-Jolma.

**Formal analysis:** Veijo Kaitala, Mikko Koivu-Jolma.

**Investigation:** Veijo Kaitala, Mikko Koivu-Jolma.

**Methodology:** Veijo Kaitala, Mikko Koivu-Jolma, Jouni Laakso.

**Software:** Veijo Kaitala, Mikko Koivu-Jolma.

**Visualization:** Veijo Kaitala, Mikko Koivu-Jolma.

**Writing – original draft:** Veijo Kaitala, Mikko Koivu-Jolma, Jouni Laakso.

**Writing – review & editing:** Veijo Kaitala, Mikko Koivu-Jolma, Jouni Laakso.

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
