## [Decision Letter · Decision Letter 0]

20 Apr 2021

PONE-D-21-07782

Infective prey leads to a partial role reversal in a predator-prey interaction

PLOS ONE

Dear Dr. Kaitala,

Thank you for submitting your manuscript to PLOS ONE. After careful consideration, we feel that it has merit but does not fully meet PLOS ONE’s publication criteria as it currently stands. Therefore, we invite you to submit a revised version of the manuscript that addresses the points raised during the review process.

My mind has been switching between “Major Revision” and “Rejection.” In the next set of reviews, I plan to invite someone who is familiar with infectious disease modeling. I encourage you to investigate infectious disease models (both SIR models and vector-born disease models). The presented model looks like that of a vector-born disease with a part of prey is infected with a pathogen. Please note that it is plausible that I might decide to reject the submission in the next round if the reviewer finds that the model already exist in the infectious disease literature. I provide more detailed comments separately.

We look forward to receiving your revised manuscript.

Kind regards,

Masami Fujiwara, PhD

Academic Editor

PLOS ONE

Journal Requirements:

Additional Editor Comments:

As reviewer 1 points out, it is not clear how representative the model is. Furthermore, you cannot stabilize a system that is already "stable". Looking at the figures, I think the system is already stable under a wide range of parameter space (including alpha=0).

You can have r_c=0 or positive as well as r_s=0 or positive, representing whether prey (or predator) is specialist or generalist. Analogous to mutualism models, we could also have k_c<0 or >0 as well as k_s<0 or >0, depending on whether they are obligate or facultative. I think investigating the dynamics of the model under each combinations of these conditions is potentially interesting (and some of them will produce “instability” without infection). -- just a suggestion for improvements.

As reviewer 2 points out, the model is also like SIR models. A large body of literature on SIR models exist. The model results should also be discussed in terms of the existing knowledge of the SIR models. I personally think the model is more like a vector-born disease model rather than an SIR model. Right now, the model is interpreted as having prey that is an energy source as well as pathogenic. However, it is more natural to interpret that the prey is carrying a pathogen (e.g. virus, bacteria, trematode, nematode, myxozoan, etc.). It may be easier to justify the model as a vector-born disease. Similarly to SIR models, there are a large body of literature on vector-born diseases. I would not be surprised if someone else already developed the model like the one presented (e.g. malaria infection model).

Reviewer 2 also raised some concerns with regard to the units of parameters. Some justifications of the parameter estimations should be presented.

Reviewers' comments:

Reviewer's Responses to Questions

**Comments to the Author**

1. Is the manuscript technically sound, and do the data support the conclusions?

Reviewer #1: Partly

Reviewer #2: Yes

2. Has the statistical analysis been performed appropriately and rigorously? 

Reviewer #1: Yes

Reviewer #2: N/A

3. Have the authors made all data underlying the findings in their manuscript fully available?

Reviewer #1: Yes

Reviewer #2: Yes

4. Is the manuscript presented in an intelligible fashion and written in standard English?

Reviewer #1: No

Reviewer #2: No

5. Review Comments to the Author

Reviewer #1: Dear authors,

your study considers how role reversal in a predator-prey system may affect the persistence of the prey and the predator species. Although the focus of your study and thus the ecological question is very interesting, I am not entirely convinced that the current formulation of the model may significantly enlarge our understanding of the potential importance of role reversal within natural predator-prey systems (for further details see my complete review attached).

Sincerely yours

Reviewer #2: While the topic of the manuscript on a food web with the possibility of reversesd prey-predator roles is quite interesting, and could as such be relevant to a number of empirical systems, I think it is not yet matured enough to be ready for publication. First of all I think that the explanation of the food web, the role of the bacteria as prey as well as infectious disease should be more clearly explained using established nomenclature. Specifically the model should not only be described as a Lotka-Volterra predator-prey model, but as a combination of a standard Lotka-Volterra predator prey and a SIR model (an established approach to model infectious disease dynamics). There is a lot of confusion on terms throughout the manuscript, see the more detailed comments directly to the article in the attached pdf.

While the explanation on your parameters seems as such to be quite detailed, they are not at all clear. Specifically it is not clear to me what is the scientific basis that support the terms used, based on wet or dry-weight, to derive specific rates like assimilation efficiencies (L134-137). Also the units used are missing a relation to volume or area. The "attack rate" of the cucumber for bacteria to me would be better described as a 'filtration rate' and not an 'attack rate', where cucumbers filter a certain volume of sediment per unit time and therefore feed on bacteria proportional to the density of the bacteria in the sediment.

Why did the authors choose a model with state variables measured in terms of individuals instead of biomass? Since the body size of cucumber versus bacteria are several orders of magnitude apart from each other, I would find it more reasonable and easier to judge on the parameterization, if state variables and parameters would be indicated on a per unit biomass basis.

The result section is quite lengthy, but still not always on the point (see detailed comments in the pdf).

The authors could more clearly relate the analytical findings to the numerical results and specifically use the analytic findings to derive some general insight on the influence of certain parameters on coexistence and dominance patterns of prey and predator. These theoretical insights should then in the discussion be related to their relevance and implications for real systems, the specific study system, as well as other systems.

The discussion is not very well developed. The authors do neither highlight the main results nor put them adequately into context of existing work. The relevance of findings from this model for other model systems is very vague and it is not clear how and why the mentioned empirical systems for example the pike and stickleback system, or agriculture could be related to the investigated model system.

6. PLOS authors have the option to publish the peer review history of their article (what does this mean?). If published, this will include your full peer review and any attached files.

Reviewer #1: No

Reviewer #2: No

---

## [Author Response · Author response to Decision Letter 0]

26 Jun 2021

We thank the editor and the referees for a thorough review process. We feel that we have been able to respond to all the comments adequately.

---

## [Decision Letter · Decision Letter 1]

23 Jul 2021

PONE-D-21-07782R1

Infective prey leads to a partial role reversal in a predator-prey interaction

PLOS ONE

Dear Dr. Kaitala,

Thank you for submitting your manuscript to PLOS ONE. After careful consideration, we feel that it has merit but does not fully meet PLOS ONE’s publication criteria as it currently stands. Therefore, we invite you to submit a revised version of the manuscript that addresses the points raised during the review process.

We received additional comments from a new reviewer. The comments agree with my assessment of the previous version (as well as the current version) of the manuscript. I would like to suggest three potential ways to mediate the issue: (1) provide a better connection with the existing literature in infectious disease models in general, (2) compare the model and results with the Nicholson-Bailey model, or (3) make the review comments available to the public. The originality of the research is one of the conditions for publication in the PLOS One, and the readers will need to know how the new knowledge is built on the existing ones. I am recommending Minor Revision because I think action (2) or (3) is straightforward. Action (1) can also be achieved quickly if you invite someone who is familiar with infectious disease models. However, it is probably important to note that Minor Revision is not the same as conditional acceptance (it is a minor in effort but still an important revision for the final decision).

We look forward to receiving your revised manuscript.

Kind regards,

Masami Fujiwara, PhD

Academic Editor

PLOS ONE

Journal Requirements:

Reviewers' comments:

Reviewer's Responses to Questions

**Comments to the Author**

1. If the authors have adequately addressed your comments raised in a previous round of review and you feel that this manuscript is now acceptable for publication, you may indicate that here to bypass the “Comments to the Author” section, enter your conflict of interest statement in the “Confidential to Editor” section, and submit your "Accept" recommendation.

Reviewer #3: (No Response)

2. Is the manuscript technically sound, and do the data support the conclusions?

Reviewer #3: Yes

3. Has the statistical analysis been performed appropriately and rigorously? 

Reviewer #3: Yes

4. Have the authors made all data underlying the findings in their manuscript fully available?

Reviewer #3: Yes

5. Is the manuscript presented in an intelligible fashion and written in standard English?

Reviewer #3: No

6. Review Comments to the Author

Reviewer #3: Review for: Infective prey leads to a partial role reversal in a predator-prey interaction

I'm going to keep this review relatively short and in keeping with Plos One guidelines. The science (methods and results) appear sound, however, as a draft, lack of context or awareness of large bodies of important work is disconcerting, and as a consequence the science doesn't add much to our knowledge of ecology. Note that this is my first review of this, I did not participate in the first review.

The authors implement a straightforward Lokta-Voltera style coupled differential equation system to model predator-prey dynamics, though not grounded in any data, and describing this system as a predator-prey system seems a bit contrived. The results are unsurprising: when a "prey" species begins to behave increasingly like a pathogen, then the "predator" (it's hard for me to call a bacteriophage a predator) tends to die and the pathogen does better, while when a "prey" is less and less pathogenic (is more and more tasty), the predator does better.

Although I think the science is basically sound (though not particularly consequential), I have some issues with the way the authors have cast the system. The idea that this is a special case of a predator-prey system is a big stretch, and I feel the system is probably much better characterized as a special case of a host-pathogen system, for which there has been a great deal of previous thought, research, and modelling work to build upon and contextualize the system with. Although I was eventually convinced that could think of V. splendidus as a consequential food source, I had to do a lot of digging in the literature -- the authors did not cite any papers the indicated if the biomass of V. splendidus was significant in the bacterial community -- something they need to set up the model as they have. Instead, I had to convince myself that V. splendidus would form a significant fraction of the bacterial biomass in this or similar systems by reading other papers on the topic. If V. splendidus is never or rarely of significant biomass, the system *definitely* has to be considered host-pathogen problem and not a predator-prey problem. So evidence that V. splendidius is a large fraction of the bacterial biomass in some relevant circumstances needs to be provided in the introduction to set up the problem, otherwise the model simply isn't well enough grounded in any real system to be helpful or informative at all.

I think more problematic is the lack of recognition, citation, or development of the model presented's relationship to two other key bodies work. The first is the enormous body of epidemiological and host-pathogen models, which are of course highly informative of the dynamics modeled here. At least as important an oversight is the classic but uncited Nicholson-Bailey model (Nicholson, A.J. and Bailey, V.A., 1935. The Balance of Animal Populations), who built a Lokta-Voltera predator-parasitoid model very, very similar to the model presented in this paper (and did so 85 years ago), and the large body of work, papers, and model extensions that grew from that classic model. None of that body of work is cited here, and it seems the authors are unaware of it. This is a problem given its similarity in both form and substance. So that context needs to be added to both the intro and discussion.

Currently the discussion, which is normally the place where authors place their findings into the context of the larger body of scientific work, has no references (which is a-typical to say the least) and mostly serves to reiterated the results. That's just not acceptable.

Finally, the authors have made minimal effort in terms of labels, colors, font sizes, line width, legend, etc. with respect to making their figures intuitive and informative.

Altogether this paper just feels sloppy, it doesn't seem to be aware of other important and relevant bodies of work and is consequently reinventing the wheel to some degree with a contorted pitch about predator-prey role reversals. It's really a host-parasite or host-pathogen problem, which have been much better worked. If this were any other journal, I would not recommend publication. Here I can only ask that at the very least the authors work to better cite and ground their work in relevant literature and rework their figures so they are of reasonable quality and readability.

7. PLOS authors have the option to publish the peer review history of their article (what does this mean?). If published, this will include your full peer review and any attached files.

Reviewer #3: No

---

## [Author Response · Author response to Decision Letter 1]

27 Aug 2021

Author Responses to the Academic Editor

Thank you very much for the second round of the referee process. 

We really appreciate the comments by the Academic Editor and the third reviewer. We have revised the MS following the advices of the Academic Editor and the the reviewer. 

In particular, we now provide a better connection with the existing literature in infectious disease models in general (Option (1) provided by the Editor). We introduce the well-established disease models preliminary in the introduction. A more detailed and lengthy comparison of the models is presented in the Discussion, where we characterize the disease model classes and point out crucial differences between them and our model. 

Addressing Option (1) also covers a comparison with the Nicholson-Bailey model as we have compared our model with the four most well-known disease model classes, one of which is the host-parasitoid model, also known as Nicholson-Bailey model (Option (2) provided by the Editor). 

Thus, we conclude that that there is always at least one crucial difference between our model and any of the other well-known disease models. Thus, we continue defend our argument about role reversal in a predator-prey interaction.

As an important part of our revision we improved our argument about the Vibrio splendidus being significant in the bacterial community and in the diet of the A japonicas (as was requested by reviewer 3).

If the current version of the MS cannot be accepted for publications in the PLOS One, we would be happy to follow the third option (3) where the review comments will be made available to the public.

Author Responses to the Reviewer

Thank you very much for the critical and competent evaluation of our MS. We agree with several issues and points raised by the reviewer in his/her report.

We made extensive revisions to improve the MS (following the justified advices of the reviewer), and also, to defend our view on the problem.

First, the reviewer states that the “evidence that V. splendidus is a large fraction of the bacterial biomass in some relevant circumstances needs to be provided in the introduction to set up the problem”.

This is an important point in order to better pose the problem. In response, we have added into the Introduction the following text on the role V. splendidus in the diet of the A. japonicus (supported with new adequate references, lines 62-68): V. splendidus is an efficient decomposer, which allows it to thrive in many environments [21,22]. To this end, Vibrios form a notable fraction of the bacterial flora in the sediment. Especially in nutrient rich areas near the range of A. japonicus Vibrios form one of the most abundant bacterial groups [22,23]. Bacteria are most abundant in the detritus, where the bacterial density can be hundredfold compared to the sea water above the detritus [20,23].

We also comment this in the Discussion (lines 480-485): Decomposer bacteria are abundant in the sediment [20,22,23], especially at the nutrient rich sea grass beds [23]. Because sea grass beds and the adjancent areas host also large numbers of detrivores and bacterivores that include sea cucumbers [23], bacteria play important part in the food web [11,18], although the details of the interactions are still poorly understood.

Thus, we continue defending our view that bacteria form an important food source for the sea cucumber, and that A. japonicus can be treated as a predator to V. splendidus.

Second, we agree that a proper comparison of our model to several well stablished disease models was lacking in our previous version of the MS. Even if our focus was in the predator-prey interaction we now feel and agree that this comparison is most important to communicate to the readers. We have now raised this issue in the introduction on a general level (lines 94-104). 

We continue comparing the similarities and differences between our model and other disease models in more detail in the Discussion (lines 521-571). In our lengthy and detailed comparison we argue that there is always at least one crucial difference between our model and any of the well-known disease models. We are not aware of any source reporting a theory for a similar species interaction with our case.

Thus, we continue to consider or study be an example of role reversal interaction in a predator-prey interaction, even if the prey happens to be a parasite.

Third, we have redrawn Figures 2-6 and the figure texts in order to improve their readability and clarity.

---

## [Editor Report · Decision Letter 2]

3 Sep 2021

Infective prey leads to a partial role reversal in a predator-prey interaction

PONE-D-21-07782R2

Dear Dr. Kaitala,

We’re pleased to inform you that your manuscript has been judged scientifically suitable for publication and will be formally accepted for publication once it meets all outstanding technical requirements.

Kind regards,

Masami Fujiwara, PhD

Academic Editor

PLOS ONE
---

## [Editor Report · Acceptance letter]

9 Sep 2021

PONE-D-21-07782R2 

Infective prey leads to a partial role reversal in a predator-prey interaction 

Dear Dr. Kaitala:

I'm pleased to inform you that your manuscript has been deemed suitable for publication in PLOS ONE. Congratulations! Your manuscript is now with our production department. 

Kind regards, 

on behalf of

Dr. Masami Fujiwara 

Academic Editor

PLOS ONE